# The gut commensal fungus, *Candida parapsilosis*, promotes high fat-diet induced obesity in mice

Shanshan Sun[1,2,3,8], Li Sun[1,4,8], Kai Wang[1,8], Shanshan Qiao[1,4], Xinyue Zhao[5], Xiaomin Hu[6], Wei Chen[7], Shuyang Zhang[5], Hantian Li[1], Huanqin Dai [1,4 ✉] & Hongwei Liu [1,4 ✉]

Gut fungi is known to play many important roles in human health regulations. Herein, we investigate the anti-obesity efficacy of the antifungal antibiotics (amphotericin B, fluconazole and 5-fluorocytosine) in the high fat diet-fed (HFD) mice. Supplementation of amphotericin B or fluconazole in water can effectively inhibit obesity and its related disorders, whereas 5-fluorocytosine exhibit little effects. The gut fungus *Candida parapsilosis* is identified as a key commensal fungus related to the diet-induced obesity by the culture-dependent method and the inoculation assay with *C. parapsilosis* in the fungi-free mice. In addition, the increase of free fatty acids in the gut due to the production of fungal lipases from *C. parapsilosis* is confirmed as one mechanism by which *C. parapsilosis* promotes obesity. The current study demonstrates the gut *C. parapsilosis* as a causal fungus for the development of diet-induced obesity in mice and highlights the therapeutic strategy targeting the gut fungi.

[1] State Key Laboratory of Mycology, Institute of Microbiology, Chinese Academy of Sciences, Beijing 100101, China. [2] School of Life Sciences, University of Science and Technology of China, Hefei, China. [3] The Second Hospital of Anhui Medical University, Hefei, China. [4] University of Chinese Academy of Sciences, Beijing 100049, China. [5] Department of Cardiology, Peking Union Medical College Hospital, Chinese Academy of Medical Science & Peking Union Medical College, Beijing 100730, China. [6] Department of Medical Research Center, State Key Laboratory of Complex Severe and Rare Diseases, Peking Union Medical College Hospital, Chinese Academy of Medical Science & Peking Union Medical College, Beijing 100730, China. [7] Department of Clinical Nutrition, Dept. of Health Medicine, Peking Union Medical College Hospital, Chinese Academy of Medical Sciences and Peking Union Medical College, Beijing, China. [8] These authors contributed equally: Shanshan Sun, Li Sun, Kai Wang. ✉email: daihq@im.ac.cn; liuhw@im.ac.cn

Obesity is a complex chronic disease accompanying with many metabolic disorders. It has become a global health issue that requires lifelong managements. According to the WHO, around 39% of adults worldwide are overweight. Yet, the pathogenesis for obesity is multifactorial and remains elusive[1]. Thus, the identification of pathogenic factors involving in the development of obesity is of great significance in clinical treatments.

In the last two decades, disrupted intestinal barrier, endotoxemia, and impaired-gut-microbiota-associated changes in intestinal metabolite have been identified as important pathogenic factors involved in the development of obesity[2,3]. More importantly, growing evidence has revealed the vital roles of gut commensal bacteria in the regulation of chronic inflammation, food intake, and energy and nutrients metabolisms, which are all closely related to obesity. For example, Enterobacter cloacae, a LPS producer in the gut, was identified as a causative bacterium in the development of obesity and insulin resistance in rodents[4]; Eubacterium hallii, a butyrate-producing bacterium of the Lachnospiraceae family, was known to improve insulin sensitivity and enhance energy expenditure in db/db mice[5]; Akkermansia muciniphila, a mucin-degrading gut bacterium, was characterized as a beneficial gut symbiont for the maintenance of metabolic homeostasis[6,7]; meanwhile, Parabacteroides distasonis was found to have metabolic benefits on the high-fat diet-induced obese mice via the activation of secondary bile acid-regulated gut-liver axis and the succinate-mediated intestinal gluconeogenesis pathway[8]. On the other hand, treatment with specific antibacterial antibiotics reduced endotoxemia to alleviate obesity in the HFD-fed mice[9].

However, as indispensable members in gut microbial communities, gut fungi and their roles in the occurrence and development of obesity are little investigated[10,11]. Recently, Tahliyah's study showed the genera Thermomyces and Saccharomyces most strongly associate with metabolic disturbance and weight gain[12]. And Borges reported a mycobiota shift towards obesity, the increased yeast in obese human individuals, and more filamentous fungi in the eutrophic human individuals[13]. In this study, we characterized the anti-obesity efficacy of the antifungal antibiotics amphotericin B (Ampho), fluconazole (Flucz), and 5-fluorocytosine (5-Fc) in high-fat diet (HFD)-induced obese mice. Amphotericin B, fluconazole, and 5-fluorocytosine were shown to have distinct antifungal profile through different action mechanisms. Supplementation of amphotericin B and fluconazole in drinking water protected the HFD-fed mice against obesity, hyperlipidemia, and hepatic steatosis. However, administration of 5-fluorocytosine failed to reduce obesity. In the fungi culture, C. parapsilosis was isolated from the feces of the vehicle-treated and the 5-fluorocytosine-treated HFD-fed mice, while this fungus was not obtained in the amphotericin B-treated or the fluconazole-treated groups. An increase of Candida parapsilosis is also shown in the overweight and obese individuals by using a culture-dependent approach[13]. Furthermore, repopulation of live C. parapsilosis in gut fungi-free mice alleviated HFD-induced obesity and related disorders, which confirmed the causal relationship between intestinal C. parapsilosis and diet-induced obesity in mice.

C. parapsilosis is well known for its great potency in secreting lipase. The lipases in the intestine perform essential roles in converting dietary triglycerides into monoglyceride and free fatty acid (FFA), contributing to fat accumulation in obese individuals[14]. Herein, we found a higher level of lipase activity in the feces of HFD-fed obese mice and C. parapsilosis-colonizing HFD-fed mice. Moreover, eradication of intestinal fungi by amphotericin B or fluconazole led to a significant decrease of lipase activity and FFA in the feces of HFD-fed mice. To confirm

the contribution of lipases secreted by C. parapsilosis to HFD-induced obesity, a lipase-negative strain of C. parapsilosis was constructed. In comparison to the wildtype strain of C. parapsilosis, repopulation of the gut fungi-free HFD-fed mice with the lipase-negative strain failed to increase body weight or obesity-related dysfunctions. This finding reveals an important role of gut mycobiota in the progression of obesity and supports the idea that the suppression of gut mycobiota or fungi-related lipases can be promising strategies to alleviate obesity and metabolic dysfunctions.

## Results

**Amphotericin B or fluconazole suppressed the obesity in high fat diet-fed (HFD) mice.** Firstly, we found an expansion of Intestinal fungi in HFD obese mice by qPCR assay (Fig. 1a). To evaluate the impact of intestinal mycobiota on obesity, amphotericin B, fluconazole, and 5-fluorocytosine dissolved in drinking water with a final concentration of $100 \, \mathrm{mg \, L^{-1}}$, $500 \, \mathrm{mg \, L^{-1}}$, and $1 \, \mathrm{g \, L^{-1}}$ were given to HFD mice. As the diet control, the standard-diet fed (SD) mice were allowed to drink water supplemented with or without amphotericin B (Fig. 1b). In comparison to the vehicle-treated HFD mice, supplementation with amphotericin B or fluconazole significantly prevented body weight gain after one-week of treatment (Fig. 1c, g). After 20 weeks, amphotericin B or fluconazole treatment significantly decreased the LEE index and fat mass in the HFD mice (Fig. 1d, e). In addition, the cumulative food intake in the amphotericin B or fluconazole-treated HFD mice was decreased to 87.8 and 91.4% of the vehicle-treated HFD mice, respectively (Fig. 1f). To our attention, there was no significant difference between the group of HFD and 5-fluorocytosine (5-Fc) treatment mice in body weight gain, food intake, fat mass, or LEE index. As to the lean mice, amphotericin B-treated SD mice exhibited little variation on the above indexes compared to the vehicle-treated SD mice, which supported a direct relationship between gut fungi and high fat diet-induced obesity.

**Amphotericin B or fluconazole alleviated obesity-related disorders in high-fat diet-fed mice.** To obesity patients, dyslipidemia and nonalcoholic fatty liver disease (NALFD) are the most commonly occurred disorders affecting long-term human health. Comparing to the hyperlipidemia in the HFD mice, amphotericin B or fluconazole supplementation effectively prevented dyslipidemia, as indicated by the lower concentrations of plasma cholesterol, low-density lipoprotein cholesterol (LDL-C), free fatty acids (FFA), and triglycerides (TG) (Fig. 2a–c). Moreover, the size of adipocytes in the white adipose tissue (WAT) of amphotericin B-treated or fluconazole-treated HFD mice was smaller than that of the vehicle-treated HFD mice (Fig. 2d). For nonalcoholic hepatic steatosis, amphotericin B and fluconazole supplementation protected the treated mice from high fat diet-induced hepatic dyslipidemia and injury, as indicated by the decreased hepatic cholesterol (TC), low-density lipoprotein cholesterol (LDL-C), free fatty acids (FFA), and triglycerides (TG) levels, the liver weight to body weight ratio, and the reduced aspartate aminotransferase (AST) activity (Fig. 2e–i). The activity of plasma alanine aminotransferase (ALT) shows no difference across the six groups (Fig. 2j). Meanwhile, high fat diet-induced hepatic macrosteatosis, hepatocyte ballooning, and fat deposition were largely inhibited by amphotericin B and fluconazole treatments, as indicated by liver sections Hematoxylin and Eosin (H&E) and Oil O red staining and microscopy observation (Fig. 2k). In contrast to amphotericin B and fluconazole, the addition of 5-fluorocytosine caused little effects on the high fat diet-induced hyperglycemia, hyperlipidemia, and hepatic steatosis. The above evidence

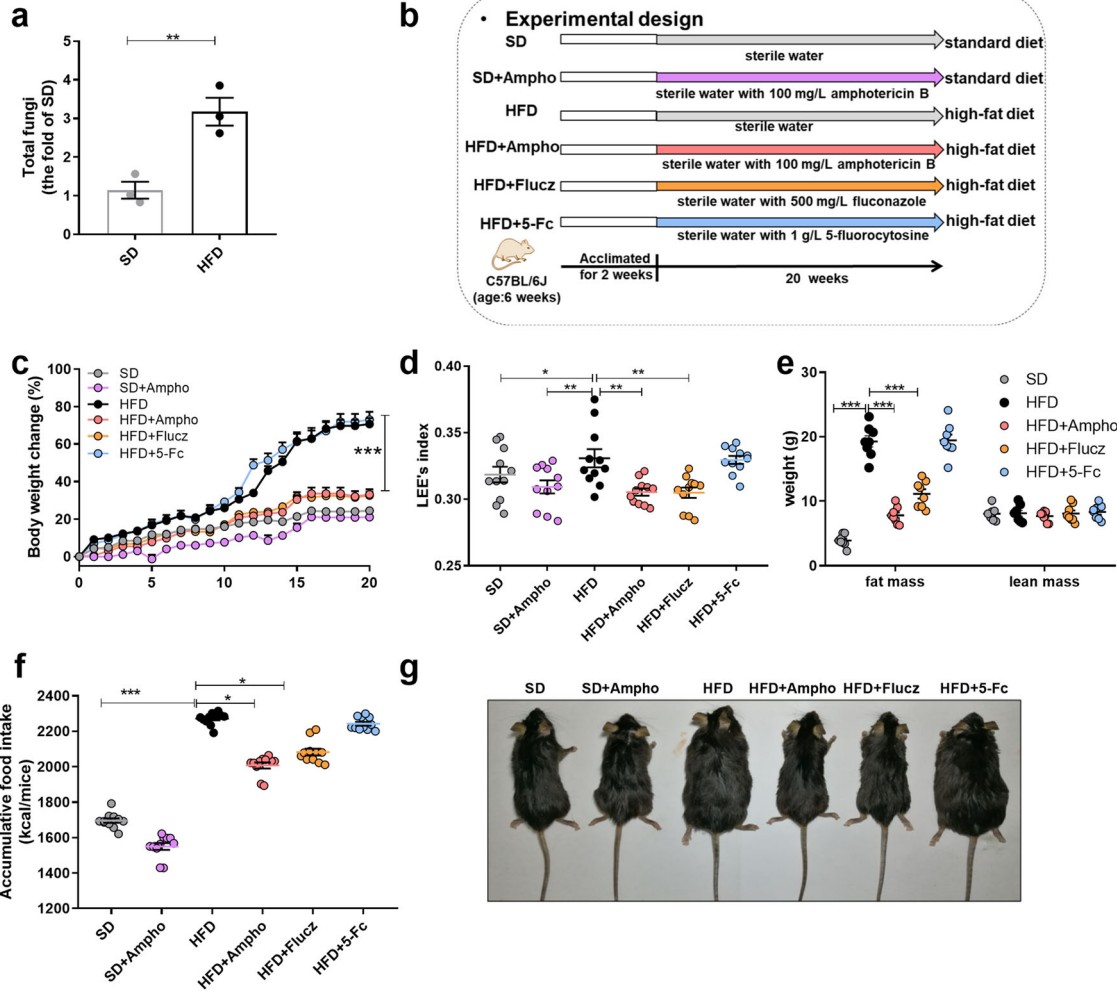

**Fig. 1 Oral administration of Amphotericin B or fluconazole suppressed the obesity in HFD-fed mice. a** Total fungi in feces were assessed by qPCR; **b** experimental design; **c** weight gain; **d** LEE's index; **e** the weight of fat mass and lean mass; **f** accumulative food intake; **g** macroscopic views of mice. SD, standard diet-fed mice group; SD + Ampho, standard diet-fed mice treated with amphotericin B; HFD, high-fat diet-fed mice group; HFD + Ampho, high-fat diet-fed mice treated with amphotericin B; HFD + Flucz, high-fat diet-fed mice treated with fluconazole; HFD + 5-Fc, high-fat diet-fed mice treated with 5-fluorocytosine. Data are presented as the mean ± SEM. **a** $N = 3$ mice per group, **c**−**f** $N = 8$−11 mice per group. Statistical analysis was performed using one-way ANOVA followed by the Tukey post hoc test for (**a**, **d**−**f**), and two-way ANOVA followed by the Bonferroni post hoc test for (**c**). *$p < 0.05$; **$p < 0.01$; ***$p < 0.001$.

confirms the contribution of gut fungi to the progression of high fat diet-induced obesity and suggests amphotericin B and fluconazole sensitive fungi as key contributors.

**Identification of obesity-associated intestinal fungus in the HFD mice.** Next, to identify intestinal fungi involved in the pathogenesis of high fat diet-induced obesity, the feces sample of mice in the above six groups were cultured on Dixon agar medium to obtain gut fungi. According to previous reports, the Dixon is the optimum solid culture medium for the isolation of gut fungi[15]. The isolated fungal strains were identified by direct internal transcribed spacer analysis. As a result, two enriched yeasts including *Candida parapsilosis* and *Naganishia globosa* were identified in the feces of the HFD obese mice (Fig. 3a). In the fecal samples of the standard diet-fed (SD) mice, amphotericin B-treated HFD-fed mice, and fluconazole-treated HFD-fed mice, no fungi were successfully cultured. Different from that of amphotericin B-treated and fluconazole-treated mice, *C. parapsilosis* were obtained in the 5-fluorocytosine-treated HFD-fed mice that showed similar obesity-related disorders as that of HFD-fed mice. Based on the above evidence, we hypothesize that

the enrichment of *C. parapsilosis* is closely associated with the occurrence and development of diet-induced obisity. Furthermore, we confirmed that *C. parapsilosis* is sensitive to both amphotericin B (MIC$_{80}$, $6.25 \pm 0.25\,\mu g\,mL^{-1}$) and fluconazole (MIC$_{80}$, $12.5 \pm 0.25\,\mu g\,mL^{-1}$), but resistant to 5-fluorocytosine (Supplementary Fig. 1 and Supplementary Table 1) by an in vitro antifungal test. Above culture experiments, together with the drug sensitiveness assay, revealed the important role of gut *C. parapsilosis* in the pathogenesis of obesity.

**Intestinal colonization of *C. parapsilosis* promoted obesity in the fungi-free HFD mice.** As a common intestinal symbiotic fungus, *C. parapsilosis* was found in the feces of overweight and obese individuals[13]. Here, we collected the feces of obese patients and healthy volunteers and detected a higher abundance of total fungi and *C. parapsilosis* in obese patients than in healthy people by qPCR (Fig. 3b, c), which is consistent with the previously reported a higher abundance of *C. parapsilosis* in obese people in Saudi Arabia[15]. However, the causal relationship between *C. parapsilosis* and obesity and related disorders remains unknown. In this study, we repopulated the amphotericin B-pretreated

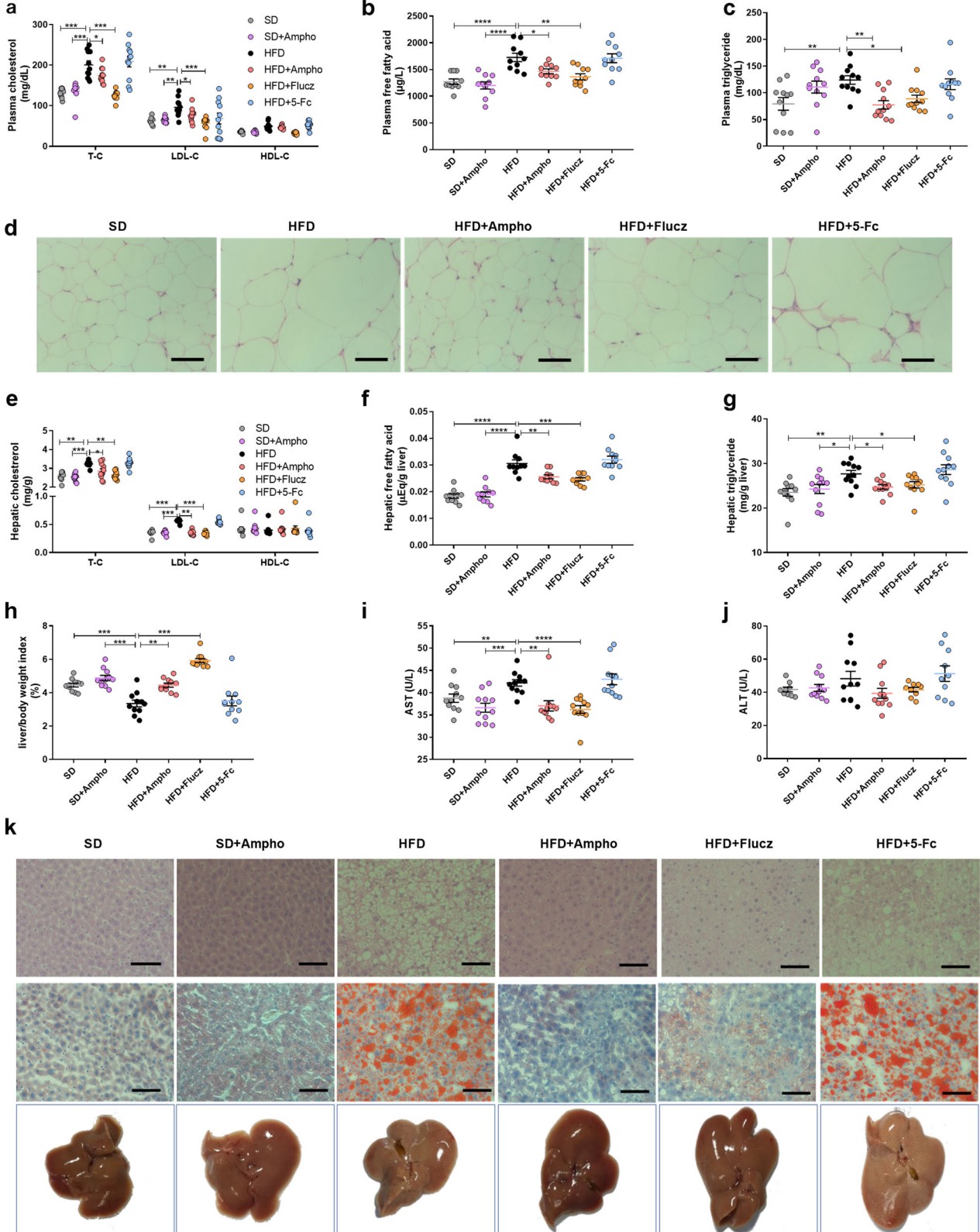

**Fig. 2 Oral administration of Amphotericin B or fluconazole alleviated obesity related disorders in HFD-fed mice. a** Plasma cholesterol; **b** plasma free fatty acid; **c** plasma triglyceride; **d** representative H&E-stained pictures of adipose tissue; **e** hepatic cholesterol; **f** hepatic free fatty acid; **g** hepatic triglyceride; **h** liver/body weight index; **i** plasma aspartate transaminase (AST); **j** plasma alanine transaminase (ALT); **k** representative H&E staining, Oil Red O staining and macroscopic views of the liver. SD, stand diet-fed mice group; SD + Ampho, stand diet-fed mice treated with amphotericin B; HFD, high-fat diet fed mice group; HFD + Ampho, high-fat diet fed mice treated with amphotericin B; HFD + Flucz, high-fat diet fed mice treated with fluconazole; HFD + 5-Fc, high-fat diet fed mice treated with 5-fluorocytosine. Data are presented as mean ± SEM. $N = 10–11$ mice per group. Statistical analysis was performed using one-way ANOVA followed by the Tukey post hoc test for (**a**–**c**), (**e**–**j**). *$p < 0.05$; **$p < 0.01$; ***$p < 0.001$; ****$p < 0.0001$.

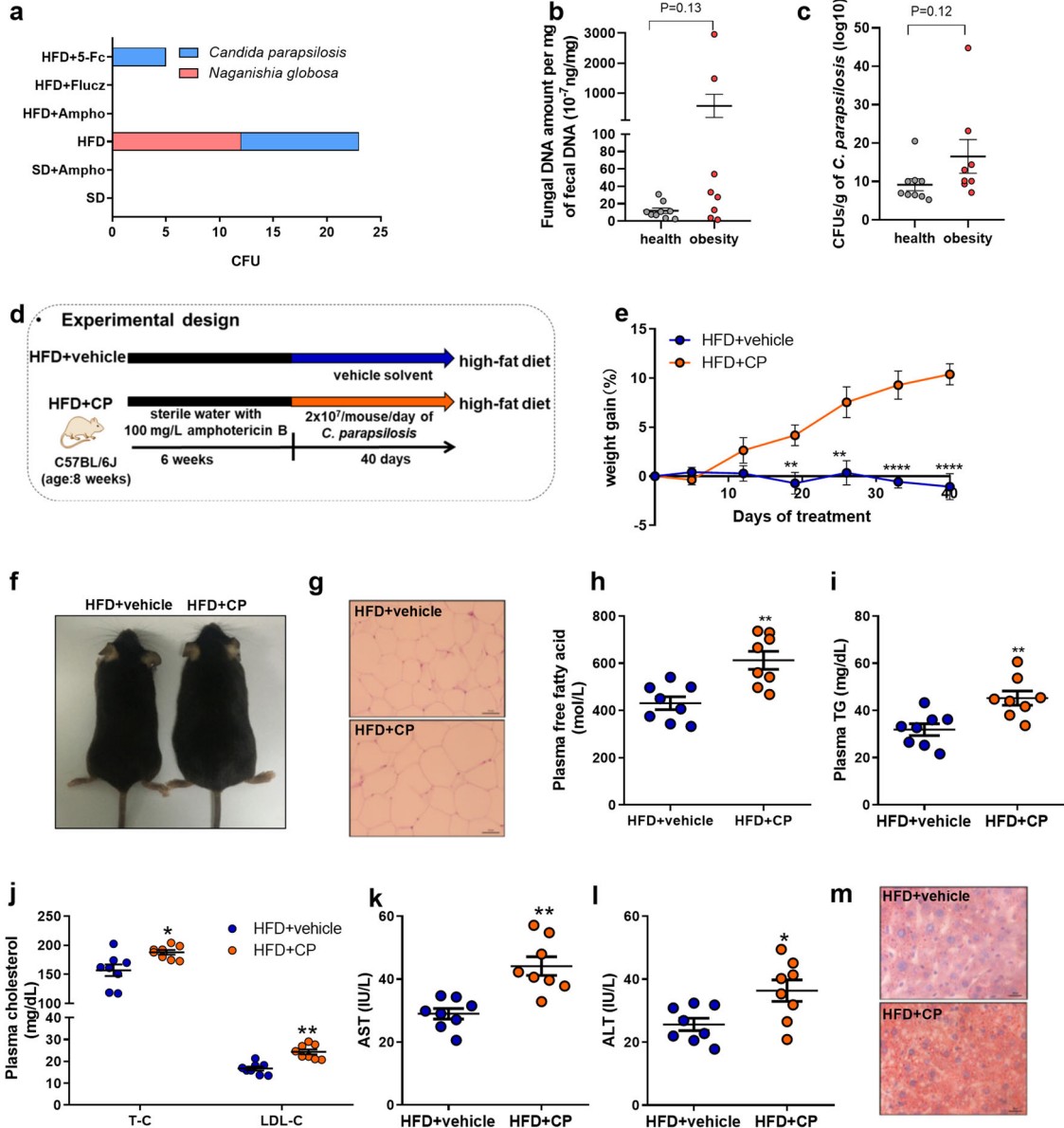

**Fig. 3 Obesity-associated intestinal fungi in HFD-fed mice. a** Fecal fungi obtained by aerobic culture on Dixon agar medium; **b** fungal DNA amount per mg of fecal DNA; **c** CFUs g$^{-1}$ of fecal C. parapsilosis; the HFD-fed mice were pretreated by addition of amphotericin B in the drinking water for six weeks, and then orally given with the live C. parapsilosis (CP) or the equal volume of PBS (vehicle) for 40 days. **d** Experimental design; **e** weight gain; **f** macroscopic views of mice; **g** representative H&E staining of adipose tissue; **h** plasma free fatty acid; **i** plasma TG; **j** plasma cholesterol; **k** plasma aspartate transaminase (AST); **l** plasma alanine transaminase (ALT); **m** representative H&E staining of the liver. HFD + vehicle, high-fat diet-fed and fungi-free mice treated with PBS; HFD + CP, high-fat diet-fed and fungi-free mice treated with the live C. parapsilosis. Data are presented as the mean ± SEM. **b, c** $N = 8-9$ human stool samples per group, **e, h−l** $N = 8$ mice per group. Statistical analysis was performed using one-way ANOVA followed by the Tukey post hoc test for (**b, c**), (**h−l**), and two-way ANOVA followed by the Bonferroni post hoc test for (E). *$p < 0.05$; **$p < 0.01$; ****$p < 0.0001$.

HFD-fed mice with C. parapsilosis to investigate the physiological functions of the gut-colonized C. parapsilosis.

The HFD-fed mice were treated with supplementation of amphotericin B in drinking water for six weeks and then were orally given live C. parapsilosis or phosphate-buffered saline (PBS) for 40 days. The anti-obesity effects of the amphotericin B pretreatment effectively lasted for the following 40 days (Fig. 3d). However, the amphotericin B-pretreated HFD-fed mice receiving live C. parapsilosis exhibited dramatic features of obesity, as indicated by the increased body weight and fat accumulation, the higher levels of plasma FFA, TG, TC, LDL-C, the enhanced activity of AST and ALT, and the increased hepatic lipid deposition (Fig. 3e−m). NO adverse effects including active state

and behavioristics were observed in mice receiving fungi. The changes in the gut microbiota of the two groups were analyzed by high-throughput sequencing (HiSeq) of the V3−V4 region of 16S rRNA genes from the caecum. Principal component analysis (PCA) showed a structural difference in the gut microbiota between the live C. parapsilosis-treated and vehicle-treated mice (Fig. 4a). Consistent with the progress of obesity, we observed a shift in gut microbiota featured with a significant increase in the phylum of Proteobacteria and Actinobacteria and the family of Desulfovibrionaceae as well as a decrease in the phylum of Firmicutes and the family of Lachinospiracea in the live C. parapsilosis-treated group by the analysis with linear discriminant analysis (LDA) effect size (LEfSe) method (Fig. 4b, c).

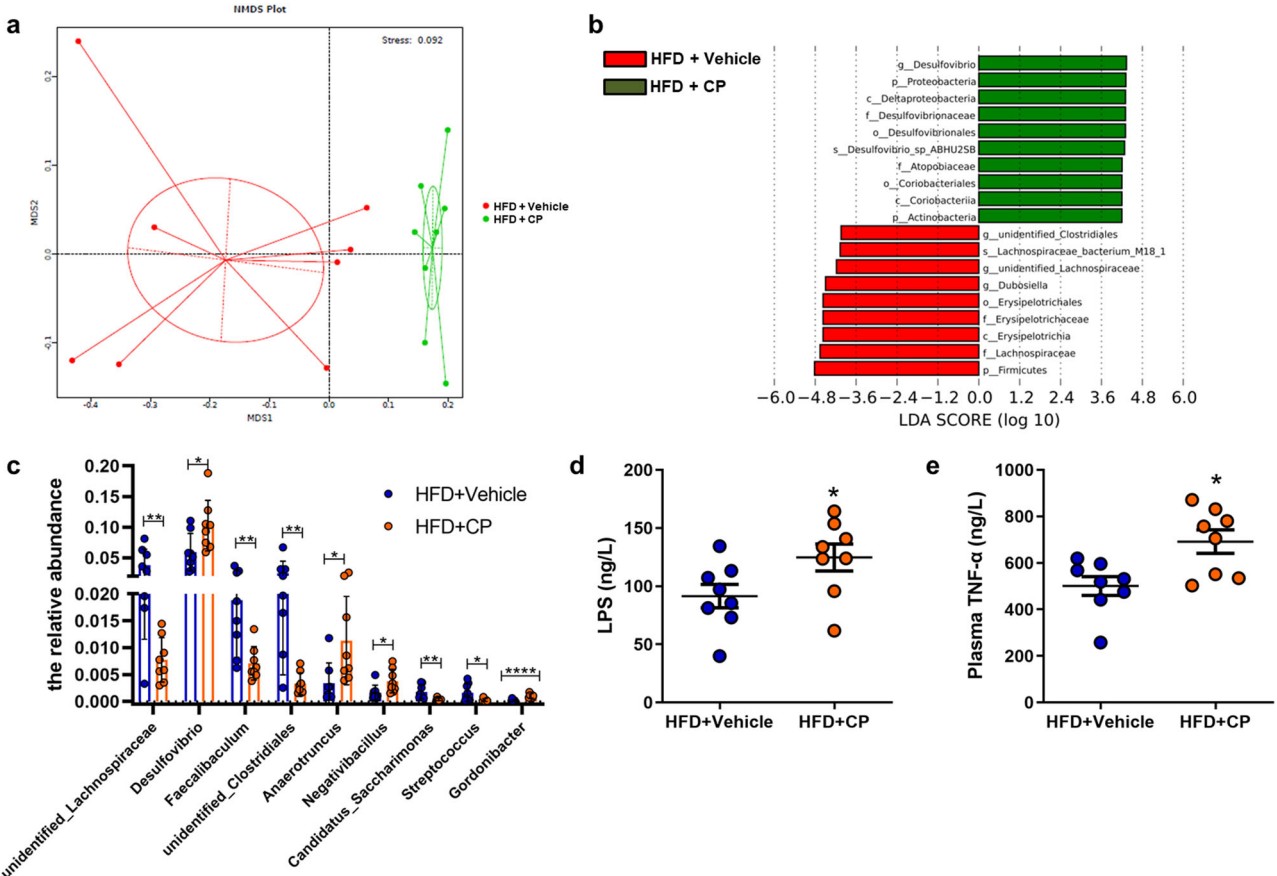

**Fig. 4 Changes of the gut bacterial community in the live *C. parpsilosis*-treated and antifungal pretreated HFD mice. a** Nonmetric multidimensional scaling (NMDS) analysis; **b** linear discriminant analysis (LDA) scores derived from LEfSe analysis; **c** T.TEST analysis at the level of genus; **d** plasma lipopolysaccharide (LPS); **e** plasma TNF-α. HFD + vehicle, high-fat diet-fed and fungi-free mice treated with PBS; HFD + CP, high-fat diet-fed and fungi-free mice treated with the live *C. parapsilosis*. Data are presented as the mean ± SEM. $N = 8$ mice per group. *$p < 0.05$; **$p < 0.01$; ****$p < 0.0001$.

Furthermore, we detected increased levels of lipopolysaccharide (LPS) and TNF-α in the plasma of live *C. parapsilosis*-innoculated HFD mice that were pretreated with amphotericin B to eradicate gut fungi (Fig. 4d, e). All the above changes demonstrated the causal relationship between intestinal *C. parapsilosis* and HFD-induced obesity.

**Lipases secreted from *C. parapsilosis* contribute to the HFD-fed induced obesity in mice**. The molecular mechanisms underlying the interaction between intestinal fungi and host are poorly understood. So far, studies have shown that components and metabolites of gut fungi mediated the interaction between intestinal fungi and host cells in alcohol-related liver disease, allergic airway inflammation, and colon cancer[16–18]. The fungus *C. parapsilosis* is well known for its strong ability to secrete lipases in the enzyme industry. The extracellular lipases are also regarded as virulence factors of *C. parapsilosis*[19]. In this study, a significant increase of the lipase activity was detected in the feces of HFD-fed mice compared to that of the SD-fed mice (Supplementary Fig. 2a). In addition, the increased lipase activity in the feces was effectively repressed by the treatment with amphotericin B or fluconazole. Repopulation of the amphotericin B-pretreated HFD mice with live *C. parapsilosis* enhanced the lipases' activity and FFA level in feces (Supplementary Fig. 2b, c). Gastrointestinal lipases that hydrolyze the triglyceride from diet into free fatty acids and monoacylglycerol are necessary for the fat accumulation in obese people. Inhibition of the activity of gastrointestinal lipases can

ameliorate obesity and its related disorders[14]. Thus, we suppose that lipases secreted from intestinal *C. parapsilosis* might contribute to the obesity of the host through the enhancement of the decomposition of triglyceride in the gut.

To further confirm the contribution of *C. parapsilosis* fungal lipases to the development of diet-induced obesity, a lipase-negative strain of *C. parapsilosis* (CP$^{KO}$) was created. As only 2 lipase genes were known in *C. parapsilosis*, we deleted the lipase locus in the *C. parapsilosis* genome consisting of adjacent genes *CpLIP1* and *CpLIP2* (Supplementary Fig. 3a, b). Lipolytic activities were examined using colonies grown on rhodamine B plates. As shown in Supplementary Fig. 3c, the WT strain exhibited strong lipolytic activities as demonstrated by the intense fluorescence under UV light; the lipase-negative mutants had no detectable activity in the plate assay. The growth and colonization ability of wild-type strain CP$^{WT}$ and the lipase-negative mutant strain CP$^{KO}$ were compared by incubation approach (Supplementary Fig. 3c, d). Knockout of *LIP1* and *LIP2* produced little effects on the growth and colonization of *C. parapsilosis* in the gut (Supplementary Fig. 4).

In the following experiments, the HFD-fed mice pretreated with amphotericin B for 2 weeks were respectively given live strain of CP$^{WT}$ and CP$^{KO}$ for another 7 weeks (Fig. 5a). In contrast to the obesity and obesity-related disorders observed in the CP$^{WT}$-treated group, the body weight gain, the size of adipose tissue, the levels of plasma TC, TG, FFA, the levels of hepatic TC and TG, the levels of AST and ALT, and the degree of hepatic steatosis in the CP$^{KO}$-treated group were little influenced (Fig. 5b–j). In addition, the lipase activity and FFA level in the

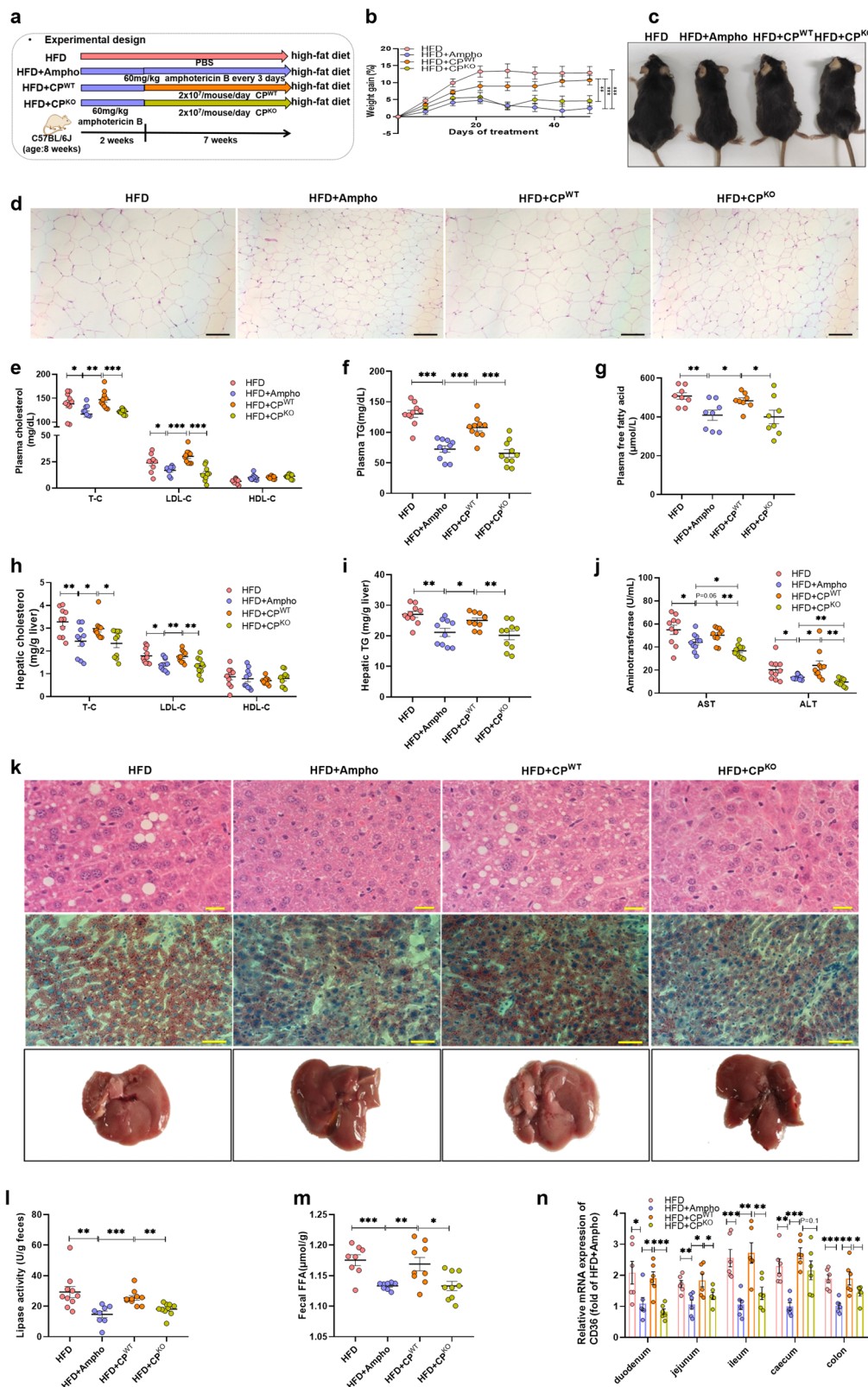

feces of mice treated with mutant strain were much lower than that of $CP^{WT}$-treated mice (Fig. 5l, m). The protein CD36 localized in the caveolae is a major player contributing for the uptake of fatty acids in the small intestine[20,21]. Next, we determined the expression of CD36 along the gut tract. It was found that the expression levels of CD36 in the epithelial cells in the HFD-fed mice treated with amphotericin B or the fungi-free

HFD-fed mice given with the lipase mutant strain of *C. parasilosis* were much lower than that of the HFD-fed mice and the fungi-free HFD-fed mice inoculated with the wild type *C. parasilosis* (Fig. 5n). The colonization of *C. parasilosis* together with the following increased CD36 in the gut strongly supported the contribution of gut fungi-derived lipases for the absorption of diet-derived fatty acids in the small intestine.

**Fig. 5 Fungal lipases from *C. parapsilosis* contribute to the HFD-induced obesity in mice.** The HFD-fed mice were pretreated by addition of amphotericin B in the drinking water for two weeks, and then orally given with the live *C. parapsilosis* (CP$^{WT}$) or the live lipase-negative mutant strain of *C. parapsilosis* (CP$^{KO}$) for 7 weeks. **a** Experimental design; **b** weight gain; **c** macroscopic views of mice; **d** representative H&E staining of the adipose tissue; **e** plasmatic cholesterol; **f** plasmatic TG; **g** plasmatic free fatty acid; **h** hepatic total cholesterol, LDL-C, and HDL-C; **i** hepatic triglyceride; **j** plasma aspartate transaminase (AST) and plasma alanine transaminase (ALT); **k** representative H&E staining and Oil Red O staining of the liver; **l** fecal lipase activity; **m** fecal FFAs level; **n** relative mRNA expression of CD36 among the intestinal tract. HFD, high-fat diet-fed mice group; HFD + Ampho, high-fat diet-fed mice treated with amphotericin B; HFD + CP$^{WT}$, high-fat diet-fed and fungi-free mice treated with live *C. parapsilosis*; HFD + CP$^{KO}$, high-fat diet-fed and fungi-free mice treated with live lipase-negative mutant strain of *C. parapsilosis*. Data are presented as the mean ± SEM. **b**, **e**−**j** and **l**, **m** $N = 8-10$ mice per group, **n** $N = 6$ mice per group. Statistical analysis was performed using one-way ANOVA followed by the Tukey post hoc test for (**e**−**j**), (**l**−**n**), and two-way ANOVA followed by the Bonferroni post hoc test for (**b**). *$p < 0.05$; **$p < 0.01$; ***$p < 0.001$.

In summary, oral treatment with two antifungal antibiotics, amphotericin B or fluconazole, effectively suppressed obesity induced by high-fat diet. The expansion of gut mycobiota was correlated with the development of HFD-induced obesity in mice, with the intestine-colonized fungus *C. parapsilosis* as a specific and causative fungus.

## Discussion

The involvement of the gut microbiota in host health and pathogenesis of obesity has been demonstrated[22]. Several studies showed that Families in the phylum Proteobacteria, such as Enterobacteriaceae and Desulfovibrionaceae that cover species producing lipopolysaccharide, are enriched in obese humans and rodents. Lipopolysaccharide, the major component of the outer membrane of Gram-negative bacteria that induces inflammation after entering the circulation, has been regarded as the major cause for the initiation of obesity[2,4,23–25]. Therapies targeting the dysregulated gut microbiota, including treatments with antibiotics against gut commensal bacteria, prebiotics, or probiotics, have been reported to make benefits to obesity and diabetes[26]. For example, administration with a mixture of antibiotics (ampicillin and neomycin) dramatically changed the composition and structure of gut microbiota and reduced endotoxemia and metabolic disorders of diabetes and obesity in both HFD-fed mice and *ob/ob* mice[3]. In another report, modulation of the gut microbiota with a mixture of norfloxacin and ampicillin was found to improve glucose tolerance in *ob/ob* mice, accompanying with the decrease of plasma lipopolysaccharides and increase of adiponectin[27]. These findings support the hypothesis that the manipulation of the obesogenic gut bacteria by a specific antibiotic might counteract weight gain.

In contrast to intensive investigations on the gut commensal bacteria, the roles of intestinal fungi associated with body health and diseases are underappreciated. Christian Hoffmann detected a total of 66 fungal genera and 184 fungal species from the feces of 96 individuals, with *Candida* as the predominant fungi[28]. The compositional changes of intestinal fungi have been found in patients with obesity, alcoholic liver diseases, gastrointestinal diseases, allergy, pancreatic cancer, and colorectal cancer[10]. As to the correlation between intestinal fungi and obesity, both the culture-dependent approach and the internal transcribed spacer-based sequencing method have shown the increase of *Candida* in the gut of obese people[13,15,29]. However, we know little about the physiological functions and underlined mechanism of intestinal fungi during the development of obesity. By using antifungal agents, we determined the causal relationship between intestinal fungi and obesity. Treatment with amphotericin B and fluconazole significantly repressed the progress of obesity, whereas the antifungal agent 5-fluorocytosine exhibited little effects on the development of obesity and related disorders. By the way, here we found there are three outliers in the HFD + 5-Fc group with very low obesity-related indicators like plasma cholesterol if compared with the rest of the HFD + 5-Fc mice, we speculate that these

mice may have obesity-resistance. Such efficacy difference indicated that a specific group of intestinal fungi sensitive to amphotericin B and fluconazole play a key role in the development of obesity. Next, we identified the fungus *C. parapsilosis* as an obesity-associated gut fungus by culturing fungal strains in feces of amphotericin B, fluconazole and 5-fluorocytosine treated mice, respectively. Further inoculation of the amphotericin B-pretreated HFD mice with live *C. parapsilosis* proved the role of *C. parapsilosis* in promoting the HFD-induced obesity. Our current investigation confirmed the causal relationship between gut *C. parapsilosis* and HFD-induced obesity in mice. In addition, the anti-obesity effect due to either the antibacterial treatment in early reports[3,28] or the antifungal treatment in this work indicates the combined effect of bacteria and fungi in the gut during the development of diet-induced obesity.

The molecular mechanisms underlying the close association between intestinal fungi and host health are receiving great attention. The enrichment of gut-derived fungal β-glucan in the circulation system of mice was demonstrated to induce liver inflammation via C-type lectin-like receptor CLEC7A on Kupffer cells, and thus contributed to alcohol-related liver disease[16]. The overgrowth of a commensal fungal *Candida* species resulting from the antibiotic-induced gut dysbiosis increased the concentration of plasmatic prostaglandin $E_2$ (PGE$_2$) that further shifted macrophage polarization in lung to promote allergic airway inflammation[17]. It was also found that the increase of gut *C. tropicalis* induced the accumulation of myeloid-derived suppressor cells to promote the development of colitis-associated colon cancer[30]. In our early work on alcoholic hepatic steatosis, an expansion of the gut fungus *Meyerozyma guilliermondii* was confirmed to aggregate the alcohol-related liver damages through gut fungi-induced PGE$_2$[31]. In this work, we demonstrated that the gut fungal lipase secreted by *C. parapsilosis* was necessary for the development of diet-induced obesity in mice, as evidenced by the fact that gut fungi-free mice gavaged with lipase-negative mutant strain of *C. parapsilosis* failed to promote diet-induced obesity as that observed under the wild type strain treatment. Fungi have been known to secret lipase for facilitating nutrient absorption from the external medium. Fungal lipases of *C. parapsilosis* are also reported as important virulence factors in the pathogen-related infections. The secreted lipases of *C. parapsilosis* facilitated the survival of the fungal pathogen in macrophages via reducing the inflammatory response of the host[32]. In the state of infection, deletion of lipase increased the inflammatory response of the host to *C. parapsilosis*. In this investigation, the similar increase in the levels of TNF-α, IL-1β, IL-6, and IL-8 was observed in the lipase-negative mutant *C. parapsilosis*-treated mice and the wild strain-treated mice, as compared with those of amphotericin B-treated HFD mice (Supplementary Fig. 5), which indicates less influence of gut fungi-secreted lipases on the host immune. Several studies showed that *Candida* can produces ligands for pattern recognition receptors (PRRs), including beta-glucans, chitin, mannans, beta-(1,2)-linked oligomannosides, and fungal nucleic acids,

which stimulate innate immune responses[33,34]. And Candida produces proinflammatory small molecular compounds like alcohol and prostaglandin[35,36]. Excessive fat accumulation is also associated with a low-grade systemic and chronic inflammatory condition[37]. Therefore, these may also be the cause of obesity caused by *C. parapsilosis*.

There are complex interactions between the intestinal microorganisms that colonize the human body. This interaction is determined by many factors, including the host's physiology, immune status, and nutritional competition. The study showed antibiotic treatment resulted in the overgrowth of a commensal fungal[38]. Another study showed Enterobacteriaceae are essential for the modulation of colitis severity by fungi[39]. To reveal the influence of *C. parapsilosis* on gut microbiota, high-throughput sequencing (HiSeq) of the V3−V4 region of 16S rRNA genes from the caecum contents was conducted. A significant increase in the phylum of Proteobacteria and Actinobacteria and the family of Desulfovibrionaceae was found in the gut microbiota of live *C. parapsilosis* group by analysis with LDA effect size (LEFSe) method, accompanying with the decrease of the phylum of Firmicutes and the family of Lachnospiracea. The result reflected *C. parapsilosis* enrichment affects the composition of intestinal bacteria. This may also be one important mechanism that *C. parapsilosis* induce obesity.

Dietary fat that is readily degraded into free fatty acids (FFA) by gastrointestinal lipases mainly contributes to the caloric extraction from food. The human gastrointestinal lipases are commonly believed to include lingual lipase, gastric lipase, and pancreatic lipase[14]. Inhibition of the digestion of dietary fat into FFAs by the lipase inhibitor Orlistat and the removal of the intestinal free fatty acids from gastrointestinal tract by some probiotic strains are validated as effective and safe strategy for the control of body weight gain[40,41]. Herein, we demonstrate the gut fungi-derived lipases to be important components of gastrointestinal lipases involving in the fat absorption, as confirmed by the lower levels of plasma and fecal FFAs in the gut fungi-free HFD mice and the lipase-negative mutant strain-treated HFD mice. Therefore, we prospect the inhibitors targeting at gut fungi-derived lipase as an alternative therapy for obesity.

The current study demonstrated that *C. parapsilosis* expansion in the gut plays a causal role in the development of obesity in mice. The obesity-promoting effect of *C. parapsilosis* is dependent on the production of fungal lipase followed by accumulation of the intestinal free fat acids. The antifungal agents, amphotericin B, and fluconazole, significantly reduced the long-term HFD-induced obesity in mice. The amphotericin B that has poor oral bioavailability and no systemic side effects deserve further clinical trials to confirm its effectiveness for the control and treatment of severe obesity. New strategies regulating gut mycobiota expansion could be searched to reduce obesity and metabolic dysfunctions.

## Methods

**Animal care and experiments**. All procedures were performed in accordance with recommendations under the Guidance for the Care and Use of Laboratory Animals of the Institute of Microbiology, Chinese Academy of Sciences (IMCAS) Ethics Committee. The protocol was approved by the Committee on the Ethics of Animal Experiments of IMCAS (Permit APIMCA2019030). All mice were housed in a 12 h light (7 AM−7 PM) and 12 h dark (7 PM−7 AM) cycle, with free access to water and chow diet. Mice were trained singly-housed in identical cages prior to data acquisition were fed with stand chow diet (Huafucang Biotech, 1021) or high-fat diet (60 kcal % fat, 20 kcal % proteins, and 20 kcal % carbohydrate, catalog D12492, Research Diets). C57BL/6J male mice were purchased from the Experimental Animal Center, Chinese Academy of Medical Sciences.

For the antifungal treatment assay on HFD-fed mice, 8-week-old C57BL/6J male mice fed with HFD were sorted into four groups ($n = 12$ each). One group was given sterile water, others were treated with sterile water supplemented with fluconazole (the final concentration is 500 mg L$^{-1}$), amphotericin B pre-solubilized in DMSO (the final concentration is 100 mg L$^{-1}$), or 5-fluorocytosine (the final concentration is 1 g L$^{-1}$), respectively. Two groups of C57BL/6J male mice fed with

standard diet (SD) were treated with sterile water or sterile water with 100 mg L$^{-1}$ amphotericin B. Treatments were continued for 20 weeks.

For *C. parapsilosis* treatment assay, *C. parapsilosis* isolated from obese mice were cultivated on potato dextrose agar for 2 days and then propagated in the potato dextrose broth (both at 28 °C) prior to oral gavage in mice. After fungal ablation with amphotericin B for six weeks, mice were divided into two groups ($n = 8$) based on their blood glucose levels and body weight. One group was given with $2 \times 10^7$ CFU mL$^{-1}$ *C. parapsilosis* daily for 40 days. The other group was treated with the vehicle solvent.

For *in vivo* assay with lipase synthesis-deficient mutant strain of *C. parapsilosis*, after fungal ablation with amphotericin B for two weeks, 30 mice were divided into three groups ($n = 10$) and then given orally with $2 \times 10^7$ CFU mL$^{-1}$ live *C. parapsilosis* (CP$^{WT}$), $2 \times 10^7$ CFU mL$^{-1}$ live lipase mutant strain of *C. parapsilosis* (CP$^{KO}$), and 60 mg kg$^{-1}$ every 3 days of amphotericin B for another 7 weeks, respectively. The HFD-fed mice receiving sterile water were used as control.

**Body weight and food Intake**. The body weight was measured every 2−3 days. Successive food consumption was calculated using the formula: [(Food mass) on Day(N) – (Food mass) on Day ($N + 1$)], and values were normalized for the number of mice per cage. The same number of mice per cage were compared between experimental groups to minimize the impact of housing density on food consumption.

**Tissue sampling**. After treatment, animals were anesthetized with diethyl ether, and blood was sampled from the portal and cava veins. After exsanguination, mice were euthanized by cervical dislocation. Subcutaneous adipose tissue deposits, intestines, and liver were precisely dissected, weighed, immediately immersed in liquid nitrogen, and stored at −80 °C for further analysis.

**Metabolic measurements and biochemical analyses**. Metabolic parameters were performed as reported earlier[42], details as follows: Levels of plasma or hepatic glucose, TC, TG, FFA, HDL-C, LDL-C, HbA1C, ALT, and AST were measured by commercial kits (Nanjing Jiancheng, Nanjing, China). Plasma Lipopolysaccharide (LPS) and TNF-α levels were quantified using LPS, TNF-α, IL-1β, IL-6, and IL-8 ELISA kit (CUSABIO, Wuhan, China). Lipase activity was measured by lipase activity assay kit (Solarbio, Beijing, China).

**Histology**. Liver and abdominal WAT were resected and fixed in 10% formalin (pH 7.4) for 4−24 h. The samples embedded in paraffin were sectioned and further stained with hematoxylin/eosin (HE staining) and Oil Red O. The slices were observed using a Zeiss Imager A2-M2 microscope (Carl Zeiss AG, Cöttingen).

**16S rRNA gene sequencing**. The caecum contents of HFD mice treated with *C. parapsilosis* or vehicle were prepared for 16S rRNA sequencing. DNA for gut microbiota analysis was subjected to 16S rRNA gene high-throughput sequencing using the IonS5TMXL platform. The 16S rRNA gene V3−V4 region was amplified using the primers F341 (CCTACGGGRSGCAGCAG) and R806 (GGAC-TACVVGGGTATCTAATC). Chimeras were filtered using USEARCH, and the remaining sequences were clustered to generate operational taxonomic units (OTUs) at 97% similarity. A representative sequence of each OTU was assigned to a taxonomic level in the RDP database using the RDP classifier. The 16S rRNA gene sequence data were processed using linear discriminant analysis effect size (LEfSe). LEfSe differences among biological groups were tested for significance using a nonparametric factorial Kruskal−Wallis sum-rank test followed by Wilcoxon rank-sum test. The 16S rRNA sequencing data were deposited in the National Microbiology Data Center (Accession numbers: NMDC40004565).

**Isolation and identification of fungi**. 100 mg of well-mixed fecal sample from five mice in each group was diluted with 900 µL phosphate buffer saline (PBS) for fungal culture. Ten-fold serial dilutions were performed. A total volume of 50 µL of each dilution was used for culturing on Dixon agar (DIX) solid culture media supplemented with antibiotics of colistin (30 mg L$^{-1}$), vancomycin (30 mg L$^{-1}$), and imipenem (30 mg L$^{-1}$), as described in early report[15]. Fungal samples were cultured at 28 °C for a week under aerobic conditions. Specimen isolation and CFU counts were conducted from the second day to the end. The identity of the isolated fungus was confirmed by Sanger sequencing of the ITS gene before use in in *vitro* experiments. Fungal genomic DNA was extracted by using the Plant DNA Extraction Kit (OMEGA, D2485). The ITS1 (TCCGTAGGTGAACCTGCGG) and ITIS4 (TCCTCCGCTTATTGATATGC) primers were used for PCR and sequencing. The PCR cycles were composed of an initial denaturating step of 94 °C for 10 min, followed by 35 amplification cycles of 94 °C for 30 s, 54 °C for 30 s, and 72 °C for 30 s and a final extension was 72 °C for 10 min.

**Collection of human stool samples**. The study sample consisted of 17 stool samples, including nine healthy people and eight obese people, with detailed information is shown in Supplementary Table 2. No enrolled subjects were given antibiotic, antifungal, or antiparasitic treatment at the time of sample collection or during the previous 2 months. Stool samples from each patient were collected and

immediately transported to the laboratory in an ice-packed container and stored at −80°C for use. Our research was approved by the Ethics Committee of Peking Union Medical College Hospital (PUMCH) and was conducted according to the principles of the Declaration of Helsinki. All individuals who participated in this study provided written informed consent, and the protocol was approved by PUMCH.

**DNA/RNA isolation and quantitative real-time PCR (qPCR) analysis**. Total RNA was extracted from the duodenum, jejunum, ileum, caecum, and colon with the TRIzol reagent according to the manufacturer's protocol (Invitrogen, Carlsbad, CA, USA). Reverse transcription was performed on 1 μg of total RNA using a cloned AMV first-strand cDNA synthesis kit (Tiangen, Beijing, China). Glyceraldehyde-3-phosphate dehydrogenase (GAPDH) was used as the house-keeping gene for normalization of the target genes expression. The qPCR mixture contained 100 ng of intestinal cDNA and 0.5 μM primers. PCR reactions were performed using Perfecta SYBR green super mix (KAPA Biosystems). PCR amplification was performed using the following cycling parameters: An initial denaturing step of 95 °C for 2 min, followed by 40 amplification cycles of 95 °C for 10 s, 59 °C for 20 s, and 72 °C for 20 s. The primer sequences are listed in Supplementary Table 3.

For isolating human fungal DNA or *C. parapsilosis*, individual fecal pellets or single colony was performed by QIAamp PowerFecal Pro DNA Kit (Qiagen). The qPCR mixture is the same as above. A standard curve using dilutions of defined content of a pure sample of single species fungal DNA was created for PCR reactions to quantify their fungal DNA content. The primer sequences are listed in Supplementary Table 3.

**Generation of lipase synthesis-deficient mutant of *Candida parapsilosis***. The SAT1-flipper method was applied to generate lipase-negative mutants from *C. parapsilosis*[43]. A 344 bp fragment containing the downstream region of the *C. parapsilosis* lipase gene was PCR-amplified from *C. parapsilosis* genomic DNA with the primers cpldown-F (5′-CCCccgccggGTACATTTCACCTTGAGTGGTC-3′) and cpldown-R (5′-CCCgagctcCTCCCAAAAAGCCATCTCAAG-3′) and then cloned into the vector pSFS2, generating the vector pDown. A 325-bp fragment containing the upstream region of the *C. parapsilosis* lipase gene was PCR-amplified with the primers cplup-F (5′-CCCgggcccTGCCCCAGTTAAACCATCACAAA-3′) and cplup-R (5′-CCCctcgagTCAATGTGGTTAAATCTGCACC-3′), then cloned into the vector pDown to construct the KO vector pCPLKO. *C. parapsilosis* strains were transformed by electroporation[44], the transformation mixture containing 40 μg *C. parapsilosis* cells and 5 μg NcoI-SacI digested and purified fragment from plasmids pCPLKO was transferred to an ice-cold electroporation cuvette (0.2 cm gap), pulsed at 1.5 kV, 25 μF, 200 Ω in a Bio-Rad electroporator. Cells were immediately suspended in YPD and incubated at 30 °C for 2 h, then plating on selection Yeast Extract Peptone Dextrose Medium (YPD) plates supplemented with 100 μg/ml Nourseothricin (Nou) and Nourseothricin-resistance (Nou$^R$) colonies were picked. Lipolytic activities were examined using colonies grown on rhodamine B plates (g L$^{-1}$ YNB, 20 ml L$^{-1}$ olive oil, 50 ml L$^{-1}$ FBS, and 1 ml L$^{-1}$ 1 M rhodamine B). Plates were subjected to UV irradiation (350 nm) and photographed.

**Growth curve measurement**. For growth curve experiment, we seeded *C. parapsilosis* (CP$^{WT}$) and lipase mutant strain of *C. parapsilosis* (CP$^{KO}$) at 10% in 25 mL of yeast extract peptone dextrose (YPD) media and incubated at 28 °C, 200 r min$^{-1}$ for 26 h. OD600 value of the fungal culture was measured every 2 h. Three replicates were prepared in two independent experiments.

**Colonization assays of CP$^{WT}$ and CP$^{KO}$**. Feces were obtained and cultured on days 1–3 post administration of *C. parapsilosis* (CP$^{WT}$) and lipase-negative *C. parapsilosis* (CP$^{KO}$). The fecal CP$^{WT}$ and CP$^{KO}$ were cultured as described above in the isolation and identification of fungi.

**Antifungal assays**. The wild-type strain of *C. parapsilosis* (CP$^{WT}$) and the lipase mutant strain of *C. parapsilosis* (CP$^{KO}$) were cultured at 28 °C for 12 h in potato dextrose broth until $1 \times 10^5$ CFU mL$^{-1}$. Then, 100 μL of CP suspension was spread on Potato dextrose agar. Amphotericin B, fluconazole, and 5-fluorocytosine were dissolved in DMSO to a final concentration of 100 μg mL$^{-1}$. Each piece of paper contained 10 μL of antibiotics. The plates were cultured at 28 °C for 2–3 days. The MIC$_{80}$ values of amphotericin B towards CP$^{WT}$ and CP$^{KO}$ were determined by the broth microdilution method in 96-well microplates at 28 °C. The cell suspensions were diluted to $1 \times 10^5$ CFU mL$^{-1}$. Amphotericin B was dissolved in DMSO to obtain final concentrations varying from 0.78 to 100 μg mL$^{-1}$. After 12–18 h, the microbial growth was assessed by measuring the optical density at 600 nm (OD600nm). Experiments were conducted in three replicates.

**Statistics and reproducibility**. All in vivo assays were performed using $n = 8-12$ mice and qPCR were performed using $n \geq 3$ independent sample. Data were analyzed using Prism (GraphPad 6.0.) and are presented as mean ± SEM. Statistical significance was determined using the unpaired two-tailed Student's t-test for single variables and two-way ANOVA for two variables. A p-value of <0.05 is considered to be statistically significant.

**Reporting summary**. Further information on research design is available in the Nature Research Reporting Summary linked to this article.

## Data availability

The 16S rRNA sequencing data were deposited in the National Microbiology Data Center (Accession numbers: NMDC40004565). Source data for the graphs and charts are provided as Supplementary Data 1. Uncropped blots are presented as Supplementary Figs. 6, 7. Any remaining information can be obtained from the corresponding author upon reasonable request.

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

## Acknowledgements

This work was financially supported by the Strategic Priority Research Program (Class B) of Chinese Academy of Sciences (grant No. XDB 38020300). The authors are very grateful to Dr. Guobo Guan for providing pSFS2 plasmid and technical support.

## Author contributions

We declare that all authors made fundamental contributions to the manuscript. S.S.S., S.L., W.K. and L.H.W. designed the research and drafted the manuscript. S.S.S., S.L., W.K., Q.S.S. and D.H.Q. conducted experiments, performed data analysis. Z.X.Y., H.X.M., C.W. and Z.S.Y. collected samples of human feces. L.H.T. contributed to revise the manuscript. All authors contributed to the interpretation of data and approved the final manuscript.

## Competing interests

The authors declare no competing interests.
