## [Peer Review File · Communications Biology]

Reviewers' comments:

Reviewer #1 (Remarks to the Author):

The authors Sun et al. have investigated the role of anti-fungal antibiotics such as amphotericin B, fluconazole and 5-fluorocytosine in gut and fungal "mycobiome" microbiome in high-fat induced obesity in mice. To do so, they have supplemented *Candida parapsilosis* to see the casual role and further, the number of *Candida parapsilosis* have reduced with antifungals with in-turn reduced obesity in mice. They found the role of *Candida parapsilosis* spp., a gut commensal fungi in induction of obesity. The manuscript is well-written and clearly discussed. That said, information on dosing strategy, high-fat diet and clinical replication may be correlated with caution, due to supplementation of either fungi or antifungals. The authors have greater potential to promote and provide clinical significance by comparing human/mice samples as they have collected both.

Comments:

1. It is a surprising not see a trace of *Candida* and *Nagashina* genera in SD. It is specific to obesity or high-fat diet supplied to the mice. There is no information about the HFD composition?
2. *C. parapsilosis* was supplemented for 40 days at 2×10^7 cfu/ml, how did the author select the dosing strategy? Can this be replicated in humans?
3. Have the authors noticed any adverse events after supplementing fungi in mice, there is no information.

Minor comments:

It will be wonderful opportunity to show:

1. Number of reads bacterial versus fungal in mice across the groups.
2. Also, the authors have collected fecal and cecal samples, it will be good to show microbial comparisons of sample type across the groups.
3. Further, the authors have human fecal samples, beneficial to to microbial comparisons between human/mice samples?
4. Number of mice per cage could have impacted due to coprophagous nature of mice?

Reviewer #2 (Remarks to the Author):

The manuscript entitled "Identification of the gut commensal *Candida parapsilosis* as a causative fungus for the development of high fat-diet induced obesity in mice" by Hongwei Liu et al. demonstrates the role of the *C. parapsilosis* produced lipase in HFD-obesity.

The manuscript is particularly interesting, well written and the conclusion are strongly supported by results. Experimental design is well organized, methods are comprehensive and sufficient to demonstrate the working hypothesis.

One major concern is related with Figure 3D: there are 6W of CP enrichment with depletion of other fungi by Amphotericin B in the drinking water followed by 40 days of CP and HFD.

What is the meaning of CP administration during the first 6 weeks if as been demonstrated that CD is susceptible to Ampho?

What is the effect of CP enrichment followed by SD? Are mice gaining more weight than CP free mice?

Minor: Figure 2C there is a substantia variability in SD diet and SD+Ampho. are plasma triglyceride more abundant when Ampho is administered to SD feed mice? SD + Ampho is missing in Figure 2K

Overall the manuscript is really well done and extremely interesting, a revision is strongly encouraged

Reviewer #3 (Remarks to the Author):

The manuscript in the title: Identification of the gut commensal *Candida parapsilosis* as a causative fungus for the development of high fat-diet induced obesity in mice is very interesting to

be accepted for publication in Journal of Communications Biology.

The experiments were well organized, performed with appropriate methods and data were clearly presented.

I would only recommend the following changes:

- This study demonstrates the gut *Candida parapsilosis* as a causal fungus for the development of diet-induced obesity in mice and you highlights the therapeutic strategy targeting the gut fungi.

How do you want to use antifungal for the therapeutic strategy in patients?

- Please show the prevalence or incidence of fungal infection or fungal including *Candida parapsilosis* in the gut of obese patients in Introduction part.

- The increase of free fatty acids (FFA) in the gut because of the production of fungal lipases from *C. parapsilosis* was confirmed as one mechanism by which *C. parapsilosis* promotes obesity. What kind of the other mechanisms that *C. parapsilosis* induce obesity?

- In result: two enriched yeasts including *Candida parapsilosis* and *Naganishia globosa* were identified in the feces of the HFD-fed mice. Why are you only interested in *C. parapsilosis* to test in mice?

- Figure 4. Please discuss the microbiome result in discussion part and also show the link between bacterial microbiota and mycobiota in your work.

- Please discuss the effects of the wild type and the lipase mutant strain *C. parapsilosis* on inflammation in HFD-induced obese mice in the discussion part.

- Please discuss the effect of plasma LPS in HFD-induced obese mice in the discussion part.

Answers to Reviewer 1

Reviewer #1 (Remarks to the Author):

The authors Sun et al. have investigated the role of anti-fungal antibiotics such as amphotericin B, fluconazole and 5-fluorocytosine in gut and fungal "mycobiome" microbiome in high-fat induced obesity in mice. To do so, they have supplemented *Candida parapsilosis* to see the casual role and further, the number of *Candida parapsilosis* have reduced with antifungals with in-turn reduced obesity in mice. They found the role of *Candida parapsilosis* spp., a gut commensal fungi in induction of obesity. The manuscript is well-written and clearly discussed. That said, information on dosing strategy, high-fat diet and clinical replication may be correlated with caution, due to supplementation of either fungi or antifungals. The authors have greater potential to promote and provide clinical significance by comparing human/mice samples as they have collected both.

Comments:

1. It is a surprising not see a trace of *Candida* and *Nagashina* genera in SD. It is specific to obesity or high-fat diet supplied to the mice. There is no information about the HFD composition?

Answers: Heisel showed that mouse fecal fungal microbiomes are not solely structured by diet, including *Candida* and *Nagashina* genera, these fungi may obtained by a variety of environmental factors¹. The low abundance in the mice given SD may make it difficult to culture under lab condition. We have added the HFD composition in part of Methods. Page25, lines 446-447.

2. *C. parapsilosis* was supplemented for 40 days at 2×10^7 cfu/ml, how did the author select the dosing strategy? Can this be replicated in humans?

Answers: We refer to the dosage of gavage *C. parapsilosis* in the literature². Whether this can be repeated in humans requires more clinical and experimental data to support.

3. Have the authors noticed any adverse events after supplementing fungi in mice, there is no information.

Answers: We have not observed any side effects after supplementing fungi in mice, including active state and behavioristics, however, the adverse events of supplementing fungi in mice need be study systematically in further study.

Minor comments:

It will be wonderful opportunity to show:

1. Number of reads bacterial versus fungal in mice across the groups.

Answers: We tended to show influence of *C. parapsilosis* supplementation on gut bacterial community in the antifungal-pretreated mice in Figure 4. To make it clear, the title of Figure 4 is changed to "Changes of gut bacterial community in the live *C. parapsilosis*-treated and antifungal pretreated HFD mice." Due to pretreatment with

amphotericin B, the number of fungal reads in the vehicle group should be trace. Thus, the number of reads bacterial versus fungal in mice can not be obtained in this work.

2. Also, the authors have collected fecal and cecal samples, it will be good to show microbial comparisons of sample type across the groups.

Answers: We only collected cecal contents of the the *C. parapsilosis* -treated mice in current study, the microbial composition can not be compared.

3. Further, the authors have human fecal samples, beneficial to to microbial comparisons between human/mice samples?

Answers: We are not able to compare the microbial composition between human and mice due to the deficiency of human samples. For reference, Yang et al. compared the fungi composition between both human and mice³. For healthy individuals, mycobiota is consist with a large proportion of genus *Candida*, along with several genus including *Pichia*, *Penicillium*, *Epicoccum*, Unclassified Fungi, *Sclerotinia* and *Stemphylium*, however, in SPF C57BL/6J mice, the gut fungi is dominant by genus *Candida*, with a proportion of *Humicola* and a lower proportion of *Fusarium*, *Sarcinomyces*, and *Aspergillus*. Above observation indicated the importance of genus *Candida* in both mice and human.

4. Number of mice per cage could have impacted due to coprophagous nature of mice?

Answers: To avoid the coprophagous nature of mice, mice were trained singly-house in identical cages prior to data acquisition, as Blacher et al described⁴.

Answers to Reviewer 2

Reviewer #2 (Remarks to the Author):

The manuscript entitled “Identification of the gut commensal *Candida parapsilosis* as a causative fungus for the development of high fat-diet induced obesity in mice” by Hongwei Liu et al. demonstrates the role of the *C. parapsilosis* produced lipase in HFD-obesity. The manuscript is particularly interesting, well written and the conclusion are strongly supported by results. Experimental design is well organized, methods are comprehensive and sufficient to demonstrate the working hypothesis.

One major concern is related with Figure 3D: there are 6W of CP enrichment with depletion of other fungi by Amphotericin B in the drinking water followed by 40 days of CP and HFD.

What is the meaning of CP administration during the first 6 weeks if as been demonstrated that CP is susceptible to Ampho?

Answers: As described in the scheme of Figure 3D, the HFD-fed mice were first treated with supplementation of amphotericin B in drinking water for six weeks to eliminate original intestinal fungus and then were orally given live *C. parapsilosis* or PBS to observe the effect of *C. parapsilosis* on obesity.

What is the effect of CP enrichment followed by SD? Are mice gaining more weight than CP free mice?

Answers: The oversupply of triglyceride in the high-fat feed are important for the development of obesity. Therefore, we did not evaluate the effect of CP enrichment followed by SD.

Minor: Figure 2C there is a substantia variability in SD diet and SD+Ampho. are plasma triglyceride more abundant when Ampho is administered to SD feed mice?

Answers: We are sorry for this mistake. The indicaiton of significance should be SD vers HFD groups. There is no significant difference in plasma triglycerides between SD and SD+Ampho groups.

SD + Ampho is missing in Figure 2K.

Answers: We have added SD+Ampho groups in Figure 2K

Overall the manuscript is really well done and extremely interesting, a revision is strongly encouraged

Answers to Reviewer 3

Reviewer #3 (Remarks to the Author):

The manuscript in the title: Identification of the gut commensal *Candida parapsilosis* as a causative fungus for the development of high fat-diet induced obesity in mice is

very interesting to be accepted for publication in Journal of Communications Biology. The experiments were well organized, performed with appropriate methods and data were clearly presented.

I would only recommend the following changes:

- This study demonstrates the gut *Candida parapsilosis* as a causal fungus for the development of diet-induced obesity in mice and you highlights the therapeutic strategy targeting the gut fungi. How do you want to use antifungal for the therapeutic strategy in patients?

Answers: The antifungal agents without gut absorption can be expected for treatment of obesity patients after deep and extensive studies in preclinical and clinical investigation. But we do not recommend antifungal treatment for patients, because the extensive use of antifungal drugs may lead to the emergence of drug-resistant strains. We believe that the best treatment is to reestablish the balance of the intestinal microbiota.

- Please show the prevalence or incidence of fungal infection or fungal including *Candida parapsilosis* in the gut of obese patients in Introduction part.

Answers: We have added the prevalence or incidence of fungal infection or fungal including *Candida parapsilosis* in the gut of obese patients in Introduction part. Page 4, lines 67-70. Recently, Tahliyah's study showed the genera *Thermomyces* and *Saccharomyces* most strongly associate with metabolic disturbance and weight gain. And Borges reported a mycobiota shift towards obesity, the increased yeast in obese human individuals, and more filamentous fungi in the eutrophic human individuals.

- The increase of free fatty acids (FFA) in the gut because of the production of fungal lipases from *C. parapsilosis* was confirmed as one mechanism by which *C. parapsilosis* promotes obesity. What kind of the other mechanisms that *C. parapsilosis* induce obesity?

Answers: We have added a brief discussion about the other mechanisms that *C. parapsilosis* induce obesity. Page 23, lines 393-399: Several study showed that *Candida* can produces ligands for pattern recognition receptors (PRRs), including beta-glucans, chitin, mannans, beta-(1,2)-linked oligomannosides, and fungal nucleic acids, which stimulate innate immune responses. And *Candida* produces proinflammatory small molecular compounds like alcohol and prostaglandin. Excessive fat accumulation is also associated with a low grade systemic and chronic inflammatory condition. Therefore, these may also be the cause of obesity caused by *C. parapsilosis*.

- In result: two enriched yeasts including *Candida parapsilosis* and *Naganishia globosa* were identified in the feces of the HFD-fed mice. Why are you only interested in *C. parapsilosis* to test in mice?

Answers: In the fecal samples of the chow diet-fed mice, amphotericin B-treated HFD-fed mice, and fluconazole-treated HFD-fed mice, no fungi were successfully

cultured. Different from that of amphotericin B-treated and fluconazole-treated mice, *C. parapsilosis* were obtained in the 5-fluorocytosine-treated HFD-fed mice that showed similar obesity-related disorders as that of HFD-fed mice. Based on above evidence, we hypothesize that the enrichment of *C. parapsilosis* is closely associated with the occurrence and development of diet-induced obesity.

- Figure 4. Please discuss the microbiome result in discussion part and also show the link between bacterial microbiota and mycobiota in your work.

Answers: Added in lines 400-413 on page 23-24. There are complex interactions between the intestinal microorganisms that colonize the human body. This interaction is determined by many factors, including the host's physiology, immune status, and nutritional competition. Study showed antibiotic treatment resulted in the overgrowth of a commensal fungal. Another study showed Enterobacteriaceae are essential for the modulation of colitis severity by fungi. To reveal the influence of *C. parapsilosis* on gut microbiota, high-throughput sequencing (HiSeq) of the V3-V4 region of 16S rRNA genes from the caecum contents was conducted. A significant increase in the phylum of Proteobacteria and Actinobacteria and the family of *Desulfovibrionaceae* was found in the gut microbiota of live *C. parapsilosis* group by analysis with linear discriminant analysis (LDA) effect size (LEFSe) method, accompanying with the decrease of the phylum of Firmicutes and the family of *Lachnospiraceae*. The result reflected *C. parapsilosis* enrichment affects the composition of intestinal bacteria. This may also one important mechanism that *C. parapsilosis* induce obesity.

- Please discuss the effects of the wild type and the lipase mutant strain *C. parapsilosis* on inflammation in HFD-induced obese mice in the discussion part.

Answers: Added in line 387-393 on page 22-23. In the state of infection, deletion of lipase increased the inflammatory response of the host to *C. parapsilosis*. In this investigation, the similar increase in the levels of TNF- α , IL-1 β , IL-6, and IL-8 was observed in the lipase-negative mutant *C. parapsilosis*-treated mice and the wild strain-treated mice, as compared with those of amphotericin B-treated HFD mice (Supplementary Fig. 5), which indicates less influence of gut fungi-secreted lipases on the host immune.

- Please discuss the effect of plasma LPS in HFD-induced obese mice in the discussion part.

Answers: Added in line 323-329 on page 20. The involvement of the gut microbiota in host health and pathogenesis of obesity has been demonstrated. Several studies showed that Families in the phylum Proteobacteria, such as *Enterobacteriaceae* and *Desulfovibrionaceae* that cover species producing LPS, are enriched in obese humans and rodents. LPS, the major component of the outer membrane of Gram-negative bacteria that induces inflammation after entering the circulation, has been regarded as the major cause for the initiation of obesity.

Reference

1. Heisel, T. et al. High-Fat Diet Changes Fungal Microbiomes and Interkingdom

Relationships in the Murine Gut. *Msphere* **2**, e00351-17 (2017).

2. Kim, YG. et al. Gut Dysbiosis Promotes M2 Macrophage Polarization and Allergic Airway Inflammation via Fungi-Induced PGE(2). *Cell Host Microbe* **15**, 95-102 (2014).

3. Yang, AM. et al. Intestinal fungi contribute to development of alcoholic liver disease. *J. Clin. Investig.* **127**, 2829-2841 (2017).

4. Blacher, E. et al. Potential roles of gut microbiome and metabolites in modulating ALS in mice. *Nature* **572**, 474-480 (2019).

REVIEWERS' COMMENTS:

Reviewer #2 (Remarks to the Author):

Authors did not substantially improve the manuscript in the present revised version, but overall it was interesting and well done even before.

In Figure 2A and 2J there are 3 outliers in the HFD+5-Fc group. Are these the same animals? If yes could you speculate about what is happening in these HFD+5-Fc mice with very low plasma cholesterol if compared with the rest of the HFD+5-Fc mice?

Reviewer #3 (Remarks to the Author):

The authors have attempted to correct all of reviewer concerns. Authors should add references to each point that you add. When you said that several reports or another study should add ref. to support.

Answers to Reviewer 2

Comment: Authors did not substantially improve the manuscript in the present revised version, but overall it was interesting and well done even before.

In Figure 2A and 2J there are 3 outliers in the HFD+5-Fc group. Are these the same animals? If yes could you speculate about what is happening in these HFD+5-Fc mice with very low plasma cholesterol if compared with the rest of the HFD+5-Fc mice?

Answers: These are the same animals; we speculate these mice may have obesity-resistance.

We had integrated the point into the discussion, Line 284: By the way, here we found there are 3 outliers in the HFD+5-Fc group with very low obesity-related indicators like plasma cholesterol if compared with the rest of the HFD+5-Fc mice, we speculate that these mice may have obesity-resistance.

Answers to Reviewer 3

Comment: The authors have attempted to correct all of reviewer concerns. Authors should add references to each point that you add. When you said that several reports or another study should add ref. to support.

Answers: We had checked that all reference in the manuscript and added references to each point that add.

- Line 254: Several studies showed that Families
- Line 326: Several studies showed that Candida can produces ligands
- Line 329: Candida produces proinflammatory small molecular
- Line 330: Excessive fat accumulation is also associated with